# Closed-Loop Transcription via Convolutional Sparse Coding

Xili Dai[1], Ke Chen[3], Shengbang Tong[2], Jingyuan Zhang[3], Xingjian Gao[2], Mingyang Li[3], Druv Pai[2], Yuexiang Zhai[2], Xiaojun Yuan[5], Heung-Yeung Shum[1,4], Lionel M. Ni[1], Yi Ma[2,3]

[1]The Hong Kong University of Science and Technology (Guangzhou), [2]University of California, Berkeley, [3]Tsinghua-Berkeley Shenzhen Institute (TBSI), [4]International Digital Economy Academy (IDEA), [5]University of Electronic Science and Technology of China

Autoencoders excel at generating models for natural images, but often lack structure and interpretability due to their use of generic deep networks. In this work, we make the explicit assumption that the image distribution is generated from a multi-stage sparse deconvolution. The corresponding inverse map, which we use as an encoder, is a multi-stage *convolution sparse coding* (CSC), with each stage obtained from unrolling an optimization algorithm for solving the corresponding (convexified) sparse coding program. Instead of directly minimizing the distributional gap between actual and generated images, we employ the closed-loop transcription (CTRL) framework to enhance the efficiency of the sparse representations. Our approach achieves comparable results on datasets like ImageNet-1K while using simpler networks and less computational power. Our method enjoys several side benefits, including more structured and interpretable representations, more stable convergence, and scalability to large datasets. Our method is arguably the *first* to demonstrate that a concatenation of multiple convolution sparse coding/decoding layers leads to an interpretable and effective autoencoder for modeling the distribution of large-scale natural image datasets.

## 1. Introduction

In recent years, deep networks have been widely used to learn generative models for real images, via popular methods such as generative adversarial networks (GAN) [20], variational autoencoders (VAE) [29], and score-matching based diffusion models [23, 24, 56]. Despite tremendous empirical successes and progress, these methods typically use empirically designed, or generic, deep networks for the encoder and decoder (or generator and discriminator, in the case of GAN). The recently proposed closed-loop transcription (CTRL) [12] framework aims to learn autoencoding models with more structured representations by maximizing the information gain, in terms of the coding rate reduction [35, 68] of the learned features. Nevertheless, like the aforementioned generative methods, CTRL uses two separate generic encoding and decoding networks which limit the potential of such a framework. We seek to remedy this issue in this work.

In image processing and computer vision, it has long been believed and advocated that sparse convolution or deconvolution is a conceptually simple and interpretable model for analyzing or synthesizing natural images. That is, natural images at different spatial scales can be explicitly modeled as being generated from a sparse superposition of a number of atoms/motifs, known as a (convolution) dictionary [65]. One conceptual benefit of such a model is that the encoding and decoding can be interpreted as mutually invertible (sparse) convolution and deconvolution processes, respectively, as illustrated in Figure 1 right. At each layer, instead of using two separate convolutional networks with independent parameters (which has been the case for most aforementioned generative or autoencoding methods), the encoding and decoding processes now share the same learned convolution dictionary. Despite their simplicity and clarity, most sparse convolution-based deep models are limited to tasks like image denoising [44] or image restoration [32]. Their empirical performance on image generation tasks has not yet been shown to competitive with the above

First Conference on Parsimony and Learning (CPAL 2024).

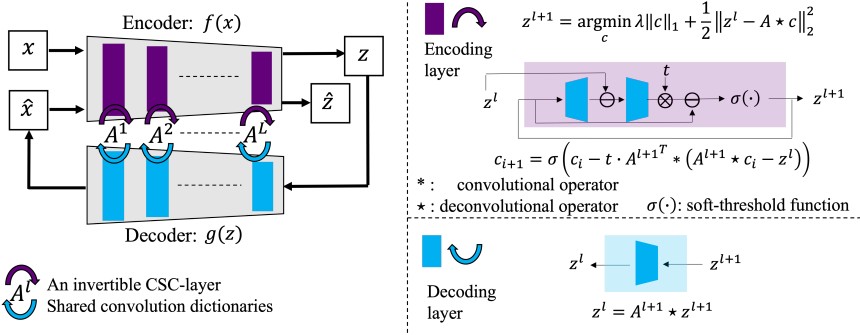

Closed-Loop Transcription | Convolution Sparse Coding (CSC) Layer

Figure 1: **Left:** A CTRL architecture with convolutional sparse coding layers in which the encoder and decoder share the same convolution dictionaries. **Right:** the encoder of each convolutional sparse coding layer is simply the unrolled optimization for convolutional sparse coding (e.g. ISTA/FISTA).

mentioned methods [1], in terms of either image quality or scalability to large datasets. Hence, in this paper, we try to investigate and resolve the following outstanding question:

> *Can we use convolutional sparse coding layers to build invertible deep autoencoding models whose performance can compete with tried-and-tested deep generative models?*

In this work, we provide an affirmative answer to this question, using invertible convolutional sparse coding layers within the CTRL framework. We improve the CTRL framework and achieve precise sample-wise alignment with the convolutional sparse coding layers. In addition, we show that deep networks constructed purely with convolutional sparse coding layers yield superior practical performance for image generation, with fewer model parameters, and less computational cost. Our work provides compelling empirical evidence which suggests that a multi-stage sparse (de)convolution has the potential to serve as an interpretable and effective model for natural image analysis and generation. To summarize, the proposed CSC based autoencoders enjoy the following benefits:

1. *Good performance on large datasets.* Compared to previous sparse coding based generative methods, our method scales well to large datasets such as ImageNet-1k, with a comparable performance than the common generative methods based on GAN or VAE, under fair experimental comparisons.

2. *Better sample-wise alignment and dataset generalizability.* The learned autoencoder achieves striking sample-wise consistency despite only optimizing alignment between distributions. We also show the generalizability of the CSC based autoencoder to unseen datasets — an autoencoder trained on CIFAR-10 can be applied to reconstruct CIFAR-100.

3. *More structured representations.* The learned feature representations for each class of images tend to have sparse low-dimensional linear structure that is amenable for conditional image generation.

4. *Higher efficiency and stability.* Our method can achieve comparable or better performance compared to other autoencoding methods, with smaller networks, smaller training batch sizes, and faster convergence. The autoencoder learned is more stable to noise than generative models based on generic networks.

Our current implementation remains rather basic, and is meant to demonstrate its simplicity. There are many aspects of the implementation which may be further improved and refined for better performance and image quality.

## 2. Connections to Related Work

**Sparse Dictionary Learning.** Inspired by neuroscience studies [45, 46], sparse coding or sparse dictionary learning (SDL) has a long history and numerous applications in modeling high-dimensional

data, especially images [2, 16, 41, 65–67]. Specifically, given a dataset $\{\boldsymbol{y}_i\}_{i=1}^n$, SDL considers the problem of learning an dictionary $\boldsymbol{A}$ such that $\boldsymbol{y}_i$ has sparse representations $\boldsymbol{y}_i \approx \boldsymbol{A}\boldsymbol{x}_i, \forall i \in [n]$ with $\boldsymbol{x}_i$ sparse. To understand the theoretical tractability of SDL, several lines of works based on $\ell^1$-norm minimization [3, 18, 19, 58, 60], $\ell^p$-norm maximization (for $p \geq 3$) [49, 55, 69, 70], and sum-of-squares methods have been proposed [4, 34, 54]. Inspired by the empirical success of SDL on tasks such as face recognition and image denoising [15, 38–41, 66], our convolutional sparse coding-based networks seek to learn a sparse representation from input images and use the learned sparse representation for image generation or autoencoding purposes.

**Sparse Modeling for Generative Models.** We are not the first to consider incorporating sparse modeling to facilitate generative tasks. To our knowledge, most existing approaches focus around using sparsity to improve GANs. For example, Mahdizadehaghdam et al. [37] exploits patch-based sparsity and takes in a pre-trained dictionary to assemble generated patches. Ganz & Elad [17] explores convolutional sparse coding in generative adversarial networks, arguing that the generator is a manifestation of the convolutional sparse coding and its multi-layered version synthesis process. Both methods have shown that using sparsity-inspired networks improves the image quality of GANs. However, these two works either use a pretrained dictionary or limit to smaller scales of data, such as the CIFAR-10 dataset. Aberdam et al. [1] uses sparse representation theory to study the inverse problem, developing a a two-layer inversion pursuit algorithm with great invertibility on datasets like MNIST. Nonetheless, most sparse-coding inspired generative frameworks have only been shown to work on smaller datasets like MNIST and CIFAR-10. In this work, we demonstrate that by incorporating convolutional sparse coding into a proper generative framework, namely CTRL, the convolutional sparse coding-based networks demonstrate striking performance on large datasets, and also have several benefits unseen by any of the previous generative methods.

# 3. Our Methods

Our goal is to learn an autoencoder from large image datasets that can achieve both distribution-wise and sample-wise autoencoding with high image quality. Our method will be based on a classic generative model for natural images: a multi-layer sparse (de)convolution model. The autoencoding will be established through learning the (de)convolution dictionaries at all layers. Such dictionaries are learned through the recent closed-loop transcription (CTRL) framework [12, 61, 62]. In particular, the coding rates of the sparse representations sought by the convolutional sparse coding are optimized during the learning process.

**Classic Autoencoding and Its Caveats.** In classic autoencoding problems, we consider a random vector $\boldsymbol{x} \in \mathbb{R}^D$ in a high-dimensional space whose distribution is typically supported on a low-dimensional submanifold $\mathcal{M}$. We seek a continuous encoding mapping $f(\cdot, \theta)$, say parameterized by $\theta$, that maps $\boldsymbol{x} \in \mathbb{R}^D$ to a compact feature vector $\boldsymbol{z} = f(\boldsymbol{x})$ in a much lower-dimensional space $\mathbb{R}^d$. In addition, we also seek an (inverse) decoding mapping $g(\cdot, \eta)$, parameterized by $\eta$, that maps the feature $\boldsymbol{z}$ back to the original data space $\mathbb{R}^D$:

$$f(\cdot, \theta) : \boldsymbol{x} \mapsto \boldsymbol{z} \in \mathbb{R}^d; \quad g(\cdot, \eta) : \boldsymbol{z} \mapsto \hat{\boldsymbol{x}} \in \mathbb{R}^D \tag{1}$$

in such a way that $\boldsymbol{x}$ and $\hat{\boldsymbol{x}} = g(f(\boldsymbol{x}))$ are "close," i.e., some distance measure $\mathcal{D}(\boldsymbol{x}, \hat{\boldsymbol{x}})$ is small.

In practice, we only have a set of $n$ samples $\boldsymbol{X} = [\boldsymbol{x}^1, \ldots, \boldsymbol{x}^n]$ of $\boldsymbol{x}$. Let $\boldsymbol{Z} = f(\boldsymbol{X}, \theta) \doteq [\boldsymbol{z}^1, \ldots, \boldsymbol{z}^n] \subset \mathbb{R}^{d \times n}$ with $\boldsymbol{z}^i = f(\boldsymbol{x}^i, \theta) \in \mathbb{R}^d$ be the set of corresponding features. Similarly let $\hat{\boldsymbol{X}} \doteq g(\boldsymbol{Z}, \eta)$ be the decoded data from the features. The overall autoencoding process can be illustrated by the following diagram:

$$\boldsymbol{X} \xrightarrow{f(\boldsymbol{x}, \theta)} \boldsymbol{Z} \xrightarrow{g(\boldsymbol{z}, \eta)} \hat{\boldsymbol{X}}. \tag{2}$$

In general, we wish that $\hat{\boldsymbol{X}}$ is close to $\boldsymbol{X}$ based on some distance measure $\mathcal{D}(\boldsymbol{X}, \hat{\boldsymbol{X}})$. In particular, we often wish that for each sample $\boldsymbol{x}^i$, the distance between $\boldsymbol{x}^i$ and $\hat{\boldsymbol{x}}^i$ is small.

However, the nature of the distribution of images is typically unknown. Historically, this has caused two fundamental difficulties associated with obtaining a good autoencoding (or generative model)

for imagery data. First, it is normally very difficult to find a principled, computable, and well-defined distance measure between the distributions of two image datasets, say $\boldsymbol{X}$ and $\hat{\boldsymbol{X}}$. This is the fundamental reason why in GAN [20], a discriminator was introduced to replace the conceptual role of such a distance; and in VAE [29], variational bounds were introduced to approximate such a distance. Second, most methods do not start with a clear generative model for images and instead adopt generic convolution neural networks for the encoder and decoder (or discriminator). Such networks do not have clear mathematical interpretations; also, it is difficult to enforce sample-wise invertibility of the networks [12].

Below, we show how both difficulties can be explicitly and effectively addressed in our approach. Our approach starts from a simple and clear model of image generation.

### 3.1. Multi-Stage Convolutional Sparse Coding and Decoding for Images

**A Generative Model for Images as Multi-Stage Sparse Deconvolutions.** We may consider an image $\boldsymbol{x}$, or its representation at any given stage of a multi-stage model, as a multi-dimensional signal $\boldsymbol{x} \in \mathbb{R}^{M \times H \times W}$ where $H, W$ are spatial dimensions and $M$ is the number of channels. We assume the image $\boldsymbol{x}$ is generated by a multi-channel sparse code $\boldsymbol{z} \in \mathbb{R}^{C \times H \times W}$ deconvolving with a multi-dimensional kernel $\boldsymbol{A} \in \mathbb{R}^{M \times C \times k \times k}$, which is referred to as a *convolution dictionary*. Here $C$ is the number of channels for $\boldsymbol{z}$ and the convolution kernel $\boldsymbol{A}$. To be more precise, we denote $\boldsymbol{z}$ as $\boldsymbol{z} \doteq (\boldsymbol{\zeta}_1, \ldots, \boldsymbol{\zeta}_C)$ where each $\boldsymbol{\zeta}_c \in \mathbb{R}^{H \times W}$ is a 2D array (presumably sparse), and denote the kernel $\boldsymbol{A}$ as

$$\boldsymbol{A} \doteq \begin{pmatrix} \boldsymbol{\alpha}_{11} & \boldsymbol{\alpha}_{12} & \boldsymbol{\alpha}_{13} & \ldots & \boldsymbol{\alpha}_{1C} \\ \boldsymbol{\alpha}_{21} & \boldsymbol{\alpha}_{22} & \boldsymbol{\alpha}_{23} & \ldots & \boldsymbol{\alpha}_{2C} \\ \vdots & \vdots & \vdots & \ddots & \vdots \\ \boldsymbol{\alpha}_{M1} & \boldsymbol{\alpha}_{M2} & \boldsymbol{\alpha}_{M3} & \ldots & \boldsymbol{\alpha}_{MC} \end{pmatrix} \in \mathbb{R}^{M \times C \times k \times k}, \tag{3}$$

where each $\boldsymbol{\alpha}_{ij} \in \mathbb{R}^{k \times k}$ is a 2D motif of size $k \times k$. Then, for each layer of the generator, also called the decoder, $g(\boldsymbol{z}, \eta)$, its output signal $\boldsymbol{x}$ is generated via the following operator $\mathcal{A}(\cdot)$ defined by deconvolving the dictionary $\boldsymbol{A}$ with the sparse code $\boldsymbol{z}$:

$$\boldsymbol{x} = \mathcal{A}(\boldsymbol{z}) + \boldsymbol{n} \doteq \sum_{c=1}^{C} \left( \boldsymbol{\alpha}_{1c} \star \boldsymbol{\zeta}_c, \ldots, \boldsymbol{\alpha}_{Mc} \star \boldsymbol{\zeta}_c \right) + \boldsymbol{n} \quad \in \mathbb{R}^{M \times H \times W}. \tag{4}$$

where $\boldsymbol{n}$ is some small isotropic Gaussian noise (modeling sampling or quantization errors etc.). For convenience, we use "$*$" and "$\star$" to denote the convolution and deconvolution operators, respectively, between any two 2D signals $(\boldsymbol{\alpha}, \boldsymbol{\zeta})$:

$$(\boldsymbol{\alpha} * \boldsymbol{\zeta})[i, j] \doteq \sum_p \sum_q \boldsymbol{\zeta}[i - p, j - q] \cdot \boldsymbol{\alpha}[p, q], \quad (\boldsymbol{\alpha} \star \boldsymbol{\zeta})[i, j] \doteq \sum_p \sum_q \boldsymbol{\zeta}[i + p, j + q] \cdot \boldsymbol{\alpha}[p, q]. \tag{5}$$

The overall decoder $g(\boldsymbol{z}, \eta)$ is a concatenation of multiple such sparse deconvolution layers and the parameters $\eta$ are the collection of learned convolution dictionaries $\boldsymbol{A}$'s (to be learned), as illustrated in Figure 1. Only batch normalization and ReLU are added between consecutive layers, to normalize the overall scale of the features and to ensure positive pixel values of the generated images. Details can be found in Appendix A.

**An Encoding Layer as Convolutional Sparse Coding.** Now, given a multi-dimensional input $\boldsymbol{x} \in \mathbb{R}^{M \times H \times W}$ sparsely generated from a (learned) convolution dictionary $\boldsymbol{A}$, the function of each layer of the encoder $f(\boldsymbol{x}, \theta)$ is to find the optimal $\boldsymbol{z}_* \in \mathbb{R}^{C \times H \times W}$ from solving the inverse problem from equation 4. Under the above sparse generative model, according to [65], we can seek the optimal sparse solution $\boldsymbol{z}$ by solving the following LASSO type optimization problem:

$$\boldsymbol{z}_* = \operatorname*{argmin}_{\boldsymbol{z}} \left\{ \lambda \|\boldsymbol{z}\|_1 + \frac{1}{2} \|\boldsymbol{x} - \mathcal{A}(\boldsymbol{z})\|_2^2 \right\} \quad \in \mathbb{R}^{C \times H \times W}. \tag{6}$$

We refer to such an implicit layer defined by equation 6 as a convolutional sparse coding layer. The reconstruction difference between $\boldsymbol{x}$ and $\mathcal{A}(\boldsymbol{z})$ is penalized by the $\ell_2$-norm of $\boldsymbol{x} - \mathcal{A}(\boldsymbol{z})$ flattened into a vector.

The optimal solution of $z$ given $\mathcal{A}$ will be a close reconstruction of $x$. Sparsity is controlled by the entry-wise $\ell_1$-norm of $z$ in the objective. $\lambda$ controls the level of desired sparsity. In this paper, we adopt the the fast iterative shrinkage thresholding algorithm (FISTA) [5] for the forward propagation. The basic iterative operation is illustrated in Figure 1. A natural benefit of the FISTA algorithm is that it leads to a network architecture that is constructed from an unrolled optimization algorithm, for which backward propagation can be carried out by auto-differentiation.

Hence, the encoder $f(x, \theta)$ is a concatenation of such convolutional sparse coding layers. Recently, the work of [11] has shown that such a convolution sparse coding network demonstrates competitive performance against popular deep networks such as the ResNet in large-scale image classification tasks. Note that in the generative setting, the operators of each layer of the encoder $f$ are determined by the same collection of convolution dictionaries $A$'s as the decoder $g$. Thus, in the autoencoding diagram in equation 2, the parameters $\theta$ of the encoder $f(x, \theta)$ and $\eta$ of the decoder $g(x, \eta)$ are determined by the same dictionaries. As we will see, this coupling brings tremendous benefits to the learned autoencoder, even besides interpretability.

## 3.2. Closed-Loop Transcription for Consistent Autoencoding

The above explicit generative model has resolved the issue regarding the structure of the the encoder and decoder for the autoencoding:[1] $X \xrightarrow{f(x,\theta)} Z \xrightarrow{g(z,\eta)} \hat{X}$. It does not yet address another difficulty mentioned above about autoencoding: how should we measure the difference between $X$ and the regenerated $\hat{X} = g(f(X, \theta), \eta)$? As we discussed earlier, it is difficult to identify the correct distance between (distributions of) images. Nevertheless, if we believe the images are sparsely generated and the sparse codes can be correctly identified through the above mappings, then it is natural to measure the distance in the learned (sparse) feature space.

The recently proposed *closed-loop transcription* (CTRL) framework proposed by [12] is designed for this purpose. The difference between $X$ and $\hat{X}$ can be measured through the distance between their corresponding features $Z$ and $\hat{Z} = f(\hat{X}, \theta)$ mapped through the same encoder:

$$X \xrightarrow{f(x,\theta)} Z \xrightarrow{g(z,\eta)} \hat{X} \xrightarrow{f(x,\theta)} \hat{Z}. \tag{7}$$

Their distance can be measured by the so-called rate reduction [14, 35, 68]: namely the difference between the rate distortion of the union of $Z$ and $\hat{Z}$ and the sum of their individual rate distortions:

$$\Delta R(Z, \hat{Z}) \doteq R(Z \cup \hat{Z}) - \frac{1}{2}(R(Z) + R(\hat{Z})). \tag{8}$$

where $R(\cdot)$ represents the rate distortion function of a distribution. In the case of $Z$ being a Gaussian distribution and for any given allowable distortion $\epsilon > 0$, $R(Z)$ can be closedly approximated by $\frac{1}{2} \log \det (I + \frac{d}{n\epsilon^2} Z Z^\top)$. Such a $\Delta R$ gives a principled distance between subspace-like Gaussian ensembles, with the property that $\Delta R(Z, \hat{Z}) = 0$ iff $\text{Cov}(Z) = \text{Cov}(\hat{Z})$ [35].

**Ensuring Self-Consistency of Autoencoding via a Sequential Game.** As shown in [12, 36, 47], one can provably learn a good autoencoding by allowing the encoder and decoder to play a sequential game: the encoder $f$ plays the role of discriminator to separate $Z$ and $\hat{Z}$ and $g$ plays as a generator to minimize the difference. This results in the following maxmin program:

$$\max_\theta \min_\eta \ \Delta R(Z(\theta), \hat{Z}(\theta, \eta)). \tag{9}$$

The program in equation 9 is somewhat limited because it only aims to align the dataset $X$ and the regenerated $\hat{X}$ at the distribution level. There is no guarantee that for each sample $x^i$ would be close to the regenerated $\hat{x}^i = g(f(x^i, \theta), \eta)$. For example, [12] shows that an input image of a car can be decoded into a horse; the so obtained autoencoding is not sample-wise consistent.

A likely reason for this to happen is because two separate networks are used for the encoder and decoder and the rate reduction objective function only minimizes error between distributions, not

---

[1]Although the effectiveness of the choice remains to be verified.

individual samples. Now notice that for the new convolutional sparse coding layers, parameters of the encoder $f$ and decoder $g$ are determined by the same convolution dictionaries $\boldsymbol{A}$. Hence the above rate reduction objective in equation 9 becomes a function of $\boldsymbol{A}$:

$$\Delta R\big(\boldsymbol{Z}(\theta(\boldsymbol{A})), \hat{\boldsymbol{Z}}(\theta(\boldsymbol{A}), \eta(\boldsymbol{A}))\big). \tag{10}$$

We can use this as a cost function to guide us to learn dictionaries $\boldsymbol{A}$ which are discriminative for the inputs and able to represent them faithfully through closed-loop transcription. To this end, for each batch of new data samples, we take one ascent step and then one descent step. The first, maximizing, step promotes a discriminative sparse encoder using only the encoder gradient, and the second, minimizing, step promotes a consistent autoencoding by using the gradients of the entire closed loop.

$$\max_{\theta(\boldsymbol{A})} \Delta R \ \text{step}: \quad \boldsymbol{A}_{k+1} = \boldsymbol{A}_k + \lambda_{\max} \frac{\partial \Delta R}{\partial \theta} \cdot \frac{\partial \theta}{\partial \boldsymbol{A}} \Big|_{\boldsymbol{A}_k}, \tag{11}$$

$$\min_{\boldsymbol{A}} \Delta R \ \text{step}: \quad \boldsymbol{A}_{k+2} = \boldsymbol{A}_{k+1} - \lambda_{\min} \Big( \frac{\partial \Delta R}{\partial \eta} \cdot \frac{\partial \eta}{\partial \boldsymbol{A}} + \frac{\partial \Delta R}{\partial \theta} \cdot \frac{\partial \theta}{\partial \boldsymbol{A}} \Big) \Big|_{\boldsymbol{A}_{k+1}}. \tag{12}$$

Empirically, we find that in the step to minimize $\Delta R$, taking the gradient as the total derivative with respect to the dictionary $\boldsymbol{A}$, i.e., using the gradients through both $\theta$ and $\eta$, converges to better results than just using the gradient through $\eta$ — see the ablation studies of Appendix B. As we will see, by sharing convolution dictionaries in the encoder and decoder, the learned autoencoder can achieve striking sample-wise consistency even though the rate reduction objective in equation 10 is meant to promote only distributional alignment.

We note that this optimization strategy is *different* from the usual techniques for maximin games [47]; this is because the encoder and decoder share the parameter $\boldsymbol{A}$. Nevertheless our alternating steps have the same conceptual effects as they do in the usual optimization strategy, i.e., alternatively maximizing the encoder's power and the consistency of the autoencoding.

To summarize, we explicitly model the distribution of natural images as being generated from a multi-stage sparse deconvolution model. This implies that the decoder $g(\cdot, \eta)$ should be a multi-layer sparse deconvolutional network. Thus, its inverse, the encoder $f(\cdot, \theta)$, should be a multi-layer convolutional sparse coding network. This means that we can well-approximate the sparse code features $\boldsymbol{z} = f(\boldsymbol{x}, \theta)$ by a mixture of Gaussians, so we can efficiently measure the distance between two such distributions by $\Delta R$ in closed-form. For the whole process, we rely on our *explicit* assumptions about the generative model to derive the overall or intermediate objectives, the overall network architecture and layer-wise operators, and the final optimization approach. This contrasts heavily with all extant approaches, e.g. GAN, VAEs, and score-based models, which do not rely on such explicit generative models and instead rely on heuristic constructions for their networks.

## 4. Experiments

We now evaluate the effectiveness of the proposed method. The main message we want to convey is that the convolutional sparse coding-based deep models can indeed scale up to large-scale datasets and regenerate high-quality images. Note that the purpose of our experiments is *not* to claim we can achieve state-of-the-art performance compared to all existing generative methods, including those that may have much larger model complexities and require arbitrary amounts of data and computational resources.[2] We compare our method with several representative categories of generative models, under fair experimental conditions: for instance, since our method uses only two simple networks, we mainly compare with methods using two networks[3], e.g., one for encoder (or generator) and one for decoder (or discriminator).

---

[2]For example, we will not compare with methods that require very large models such as Big-GAN [6] or NSCN++ [57].

[3]Hence, we will not compare with methods that require multiple networks for additional discriminators such as the VAE-GAN [48] and the Style-GAN [27].

| Method | Cifar-10 | | STL-10 | | ImageNet | |
|---|---|---|---|---|---|---|
| | IS↑ | FID↓ | IS↑ | FID↓ | IS↑ | FID↓ |
| *GANs* | - | - | - | - | - | - |
| DCGAN | 6.6 | 35.3 | 7.8 | - | - | - |
| SNGAN | 7.4 | 29.3 | 9.1 | 40.1 | 7.3 | 48.7 |
| *VAEs* | - | - | - | - | - | - |
| VAE | 5.2 | 55.9 | - | - | - | - |
| NVAE | - | 50.8 | - | - | - | - |
| *Flows* | - | - | - | - | - | - |
| GLOW | - | 46.9 | - | - | - | - |
| R-Flow | - | 50.8 | - | - | - | - |
| *CTRLs* | - | - | - | - | - | - |
| CTRL | 8.1 | 19.6 | 8.4 | 38.6 | 7.7 | 46.9 |
| CSC-CTRL (ours) | 8.9 | 28.9 | 9.1 | 48.1 | 12.5 | 34.5 |

Table 1: Comparison on CIFAR-10, STL-10, and ImageNet-1K. The network architectures used in CSC-CTRL are 4-layers for CIFAR-10, 5-layers for STL-10 and ImageNet respectively which are much smaller than other compared methods.

**Datasets and Experiment Setting.** We test the performance of our method on CIFAR-10 [31], STL-10 [9] and ImageNet-1k [13] datasets. The detailed implementation settings and network parameters can be found in Appendix A.1 for CIFAR-10, STL-10 and ImageNet-1k.

## 4.1. Performance on Generative Image Autoencoding

We adopt the standard FID [22] and Inception Score (IS) [53] to evaluate the generative quality of learned representations. We compare our method to the most representative methods from the following categories: GAN, VAE, flow-based, and CTRL, under the same experimental conditions – except that our method typically uses simpler and smaller models.

On medium-size datasets such as CIFAR-10, we observe in Table 1 that, in terms of these metrics, our method achieves comparable or better performance compared to typical GAN, flow-based and VAE methods, and better IS than CTRL and VAE-based methods, which conceptually are the closest to our method. Comparing to CTRL, Figure 2 showcases the different reconstructed image between CTRL and CSC-CTRL. It is clear that CSC-CTRL not only enjoys better visual quality, but also achieves much better sample-wise alignment. Visually, Figure 3 further shows an amazing sample-wise alignment between input $X$ and reconstructed $\hat{X}$ despite our method not enforcing sample-wise constraints or pixel-level loss functions!

On larger-scale datasets such as ImageNet-1k, Table 1 shows that we outperform many existing methods in Inception Score. Figure 3 shows that the decoded $\hat{X}$ looks almost identical to the original $X$, even in tiny details. All of the images displayed are randomly chosen without cherry-picking. Due to page limitations, we place more results on ImageNet and STL-10 in Appendix C.

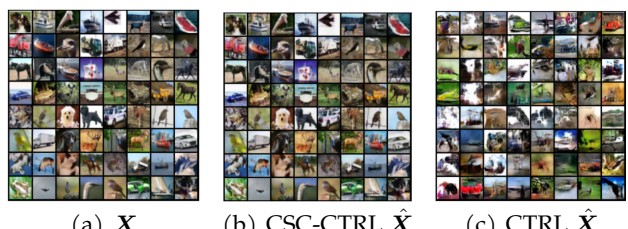

(a) $X$          (b) CSC-CTRL $\hat{X}$          (c) CTRL $\hat{X}$

Figure 2: Visualizing the auto-encoding property of the learned CSC-CTRL ($\hat{X} = g \circ f(X)$) comparing to CTRL on CIFAR-10. (Images are randomly chosen.)

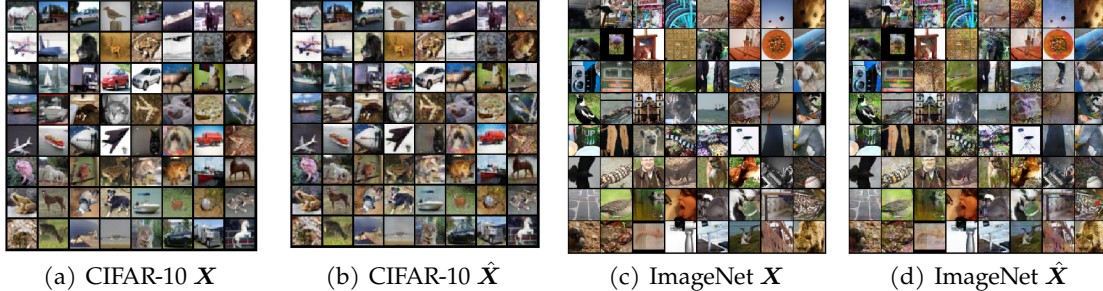

| (a) CIFAR-10 $X$ | (b) CIFAR-10 $\hat{X}$ | (c) ImageNet $X$ | (d) ImageNet $\hat{X}$ |

Figure 3: Visualizing the auto-encoding property of the learned CSC-CTRL ($\hat{X} = g \circ f(X)$) on CIFAR-10 and ImageNet. (Images are randomly chosen.)

## 4.2. Structures of Learned Representations

To evaluate the structural properties of the learned feature space, we visualize the reconstructed samples along different principal components in the feature space of learned classes. We follow the procedure done in [12], calculating the principal components of the representations in each learned class, and then reconstructing the samples with representation closest to these principal components. Each row in Figure 4 displays objects of one class; each block of 5 images shows one principal component within each class. It clearly demonstrates that we may express the image diversity within each class by simply computing the principal components of the class. Even though our method does not use class label information, the model preserves statistical diversities between classes and within each class. We provide additional generated images, feature space interpolation and cosine similarity heatmap of learned representations in Appendix C, D.

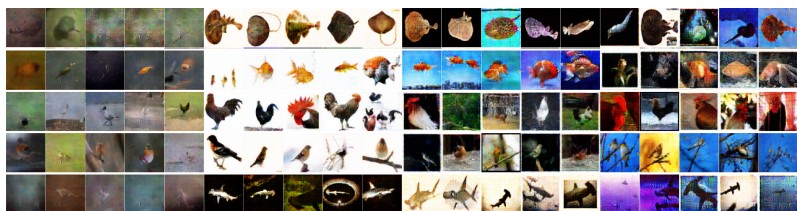

Figure 4: Five reconstructed $\hat{x} = g(z)$ from $z$'s with the closest distance to (top-4) principal components of learned features for ImageNet (class "rajidae", "goldfish", "chicken", "bird", "shark").

## 4.3. Generalizability to Autoencoding Unseen Datasets

To evaluate the generalizability of the learned model, we reconstruct samples of CIFAR-100 using a CSC-CTRL model which is only trained on CIFAR-10. Figure 5 shows a randomly reconstructed sample without cherry-picking. We observe that a lot of classes — for example, "lion", "wolf", and "snake" — which never appeared in CIFAR-10 can still be reconstructed, with high image quality. Moreover, if we visualize the samples along different principal components within the class, we see that even the variance in the out-of-domain data samples may be captured by computing the principal components. It demonstrates that our model not only generalizes image reconstruction well to out-of-domain data, but also encodes a meaningful representation that preserves diversity between and within out-of-domain classes.

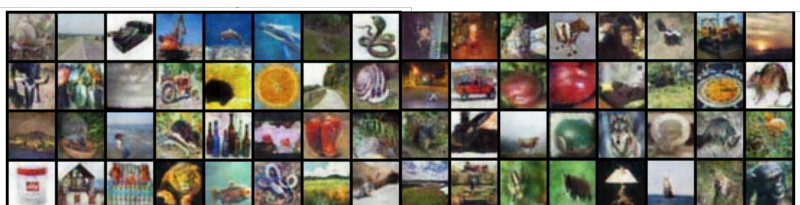

Figure 5: Visualization of randomly chosen reconstructed samples $\hat{X}$ of CIFAR-100. The autoencoding model is only trained on the CIFAR-10 dataset.

### 4.4. Stability of CSC-CTRL

To test the stability of our method to input perturbation, we add Gaussian noise with mean of 0 to the original CIFAR-10 dataset. We use $\sigma$ to control the standard deviation of the Gaussian noise, i.e., the level of perturbation. The property of the convolutional sparse coding layer makes possible a stable recovery of the sparse signals with respect to input noise and, therefore, enables denoising [15, 65]. Hence, CSC-CTRL's autoencoding also functions as denoising of noisy data. We conducted experiments on CIFAR-10, with $\sigma = 0.5$, and STL-10, with a $\sigma = 0.1$. Because CIFAR-10 has a smaller resolution, we use a larger $\sigma$ so we can visualize the noise more clearly. From Figure 6, we see that CSC-CTRL outputs a better-denoised image. When noise level are larger, CSC-CTRL has an obvious advantage over CTRL, which uses traditional convolutional layers. We also present more quantitative analysis of denoising in Appendix E.1.

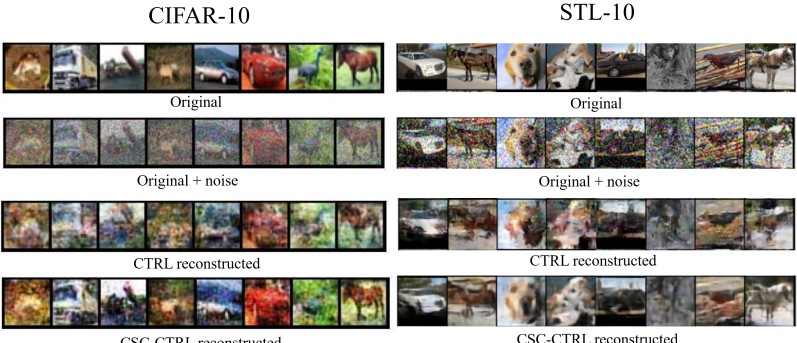

Figure 6: Denoising using CTRL and CSC-CTRL on CIFAR-10 with $\sigma = 0.5$ and STL-10 with $\sigma = 0.1$.

The CSC-CTRL model also demonstrates better stability in training than CTRL model. The coupling between the encoder and decoder makes the training more stable. For instance, the IS score of the CSC-CTRL model typically gradually increases and converges during training, whereas the CTRL model's IS score continuously drops after convergence. In addition, CSC-CTRL can converge with a wide range of batch sizes, from as small as 10 to as large as 2048, whereas CTRL can only converges with batch size larger than 512. These two properties are highly important from the perspective of engineering models within the CTRL framework. More details can be found in Appendix F.

## 5. Conclusion and Future Work

In this work, we have shown the classic and basic convolution sparse coding models are sufficient to construct strikingly good autoencoders for large sets of natural images. This leads to a simplifying and interpretable framework for learning and understanding the statistics of natural images. This new framework integrates intermediate goals of seeking compact sparse representations with an end goal of obtaining an information-rich and yet compact & structured representation, measured by the coding rate reduction. The learned models have demonstrated unprecedented generalizability and stability. We believe this gives a new powerful family of generative/autoencoding models that can better support a wide range of applications that require more interpretable and controllable image generation and understanding.

Notice that our current implementation is extremely basic and simple. There are rich variants to the convolution sparse coding layers that could promote different types of sparsity or low-dimensional structure in images [65] as well as variants to the rate reduction objective for the final feature representations [12] that could incorporate richer class-wise or sample-wise information. These variants have not been considered in this work. Hence there is ample reason to believe that the performance, scalability, and efficiency of this method can be significantly improved in the future with better engineering and implementation.

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

# A. Appendix

## A.1. Experiment Settings and Implementation Details

**Network backbones.** For CIFAR-10, we follow the 4-layers architecture which is used for MNIST in [12], replacing all the standard convolutional layers with our drop-in convolutional sparse coding layers in Table 2 and 3 without extra modifications. Similarly, we adopt the 5-layers architecture for STL-10 (see Table 4 and 5) and ImageNet-1k (see Table 4 and 5).

| $z \in \mathbb{R}^{1 \times 1 \times 512}$ |
| --- |
| $4 \times 4$, stride=1, pad=0 CSC-inv BN 256 ReLU |
| $4 \times 4$, stride=2, pad=1 CSC-inv BN 128 ReLU |
| $4 \times 4$, stride=2, pad=1 CSC-inv BN 64 ReLU |
| $4 \times 4$, stride=2, pad=1 CSC-inv 3 Tanh |

Table 2: Decoder for CIFAR-10.

| RGB image $x \in \mathbb{R}^{32 \times 32 \times 3}$ |
| --- |
| $4 \times 4$, stride=2, pad=1 CSC 64 lReLU |
| $4 \times 4$, stride=2, pad=1 CSC BN 128 lReLU |
| $4 \times 4$, stride=2, pad=1 CSC BN 256 lReLU |
| $4 \times 4$, stride=1, pad=0 CSC 512 |

Table 3: Encoder for CIFAR-10.

| $z \in \mathbb{R}^{1 \times 1 \times 1024}$ |
| --- |
| $4 \times 4$, stride=1, pad=0 CSC-inv BN 512 ReLU |
| $4 \times 4$, stride=2, pad=0 CSC-inv BN 256 ReLU |
| $4 \times 4$, stride=2, pad=1 CSC-inv BN 128 ReLU |
| $4 \times 4$, stride=2, pad=1 CSC-inv BN 64 ReLU |
| $4 \times 4$, stride=2, pad=1 CSC-inv 3 Tanh |

Table 4: Decoder for STL-10 and ImageNet-1k.

| RGB image $x \in \mathbb{R}^{64 \times 64 \times 3}$ |
| --- |
| $4 \times 4$, stride=2, pad=1 CSC 64 lReLU |
| $4 \times 4$, stride=2, pad=1 CSC BN 128 lReLU |
| $4 \times 4$, stride=2, pad=1 CSC BN 256 lReLU |
| $4 \times 4$, stride=2, pad=0 CSC 512 |
| $4 \times 4$, stride=1, pad=0 CSC 1024 |

Table 5: Encoder for STL-10 and ImageNet-1k.

## A.2. Optimization and training details.

**General Settings.** Adam [28] is adopted as the optimizer for all of our experiments. The hyper-parameters of Adam and the learning rate for each dataset will be discussed later in their own section. We choose $\epsilon^2 = 0.5$ for the maximin program (10) in all experiments, and the $\lambda$ inside the convolutional sparse coding layer is set to be 0.01 by default. For alternating minimizing and maximizing the objectives, we use the simple gradient descent-ascent algorithm. Most experiments are conducted on RTX 3090 GPUs.

**CIFAR-10.** For CIFAR-10, the learning rate is set to be $2 \times 10^{-4}$ with no decay, and we choose $\beta_1 = 0$, $\beta_2 = 0.9$ for Adam optimizer. Besides, we run 1000 epochs with mini-batch size 2000 for each experiment. In most cases, the model converges after about 300 epochs, with consistent visual quality and stable Inception Score.

**STL-10.** For STL-10, images are firstly resized to 64x64 using bilinear interpolation, and we run 1000 epochs with mini-batch size 1024, learning rate $2 \times 10^{-4}$ with no decay, and hyper-parameters $\beta_1 = 0.5$, $\beta_2 = 0.9$ for Adam optimizer. The model converges after about 300 epochs, with consistent visual quality and stable Inception Score.

**ImageNet-1k.** For ImageNet-1k, images are firstly center-cropped to 224x224 and then resized to 64x64 using bilinear interpolation during training. We run 10000 iterations with mini-batch size 128, learning rate $1 \times 10^{-4}$ and hyper-parameters $\beta_1 = 0.5$, $\beta_2 = 0.9$ for the Adam optimizer.

## A.3. Connections to Score-Matching and Diffusion

In this subsection we discuss high-level conceptual connections, both similarities and differences, between our overall methodology and the popular technique of score-matching [25, 57, 64], as well as the recent popular diffusion models [23, 57].

### A.3.1. Score Matching

Our formulation in terms of rate distortion is conceptually similar to *score matching* [25, 57, 64]. The score function is the gradient of the (un-normalized) log-likelihood, that is, $\nabla_{\boldsymbol{x}} \log p(\boldsymbol{x})$, whereas the expectation of the (negative) log-likelihood can be interpreted as the coding rate of the distribution ([10], Chapter 14). In the case of a distribution with degenerate (low-dimensional) supports, its density or log-likelihood (hence score function) is not well-defined. A natural surrogate is its (lossy) rate distortion $R(\boldsymbol{x})$ subject to a prescribed quantization error [35], which is well-defined even in these contexts. Thus, the rate distortion formulation extends the log-likelihood to high-dimensional (image) distributions with degenerate structures[4]. Hence the gradient of the rate distortion can be viewed as a surrogate to the score function.[5] In the case the distribution is (locally) approximated as a mixture of Gaussians, the score function can be efficiently computed in closed-form as the gradient of the Gaussian rate distortions.[6] The above rate reduction gives a closed-form formula for the distance between such two distributions.

### A.3.2. Diffusion and Denoising

Each sparse deconvolution layer transforms its sparse input $\boldsymbol{z}$ into a denser representation $\mathcal{A}(\boldsymbol{z})$. One can view the concatenation of these layers, i.e., the decoder $g(\boldsymbol{z}, \eta)$, as carrying out multiple steps of an incremental deformation from a sparse code $\boldsymbol{z}$ to a dense and higher-dimensional $\boldsymbol{x}$. This is conceptually analogous to the so-called *diffusion* process [23, 56, 57]. The main difference is that traditional diffusion models start at the structured high-dimensional image data $\boldsymbol{x}$ and, by incrementally adding isotropic Gaussian noise, diffuse to standard Gaussian noise. In contrast, we start with *even more organized* and lower-dimensional sparse codes $\boldsymbol{z}$ and, by adding *anisotropic* Gaussian noise (i.e. isotropic Gaussian noise pushed through the sparse deconvolution at each layer), diffuse it to get $\boldsymbol{x}$. In this way, one may consider the decoder as conducting a form of "*structured diffusion*" or "*anisotropic diffusion.*"

In the inverse direction, each convolutional sparse coding layer extracts a sparse code $\boldsymbol{z}$ from a dense representation or image via LASSO regression against a learned convolutional dictionary of the distribution of natural images. Dictionary learning is in fact one of the purposes why the score matching was introduced by [25] in the first place. Thus one can consider our encoder, which is a concatenation of such layers, as carrying out multiple steps of an incremental deformation from dense and high-dimensional image data $\boldsymbol{x}$ to a sparse and lower-dimensional encoding $\boldsymbol{z}$. This is conceptually analogous to the so-called *Langevin dynamics* process which is used in diffusion models [23, 56, 57]. The main difference between Langevin dynamics and our process is that the Langevin dynamics starts with standard Gaussian noise, and incrementally *denoises* to transform it to structured and high-dimensional natural images $\boldsymbol{x}$. In contrast, we start with the structured natural images $\boldsymbol{x}$ and incrementally "*denoise*" (regress against the learned convolutional structures) to transform it to *even more structured* output, i.e., the compact codes $\boldsymbol{z}$. In this way, one may consider the encoder as conducting a form of "*structured denoising.*"

To summarize, while diffusion models map to and from unstructured noise, our process maps to and from an explicit structure modeled by a learned sparse (de)convolution as illustrated in Figure 7.

---

[4]This is crucial if one wants to identify such structures explicitly, which is precisely the purpose of this work. Note that this differs from almost all extant generative models.

[5]Geometrically, the score function, and hence the gradient of the coding rate, indicates directions in which the encoder can most effectively compress or expand the (local) volume of the representations, measured in terms of decreasing or increasing the coding rate.

[6]Such gradients are the basic layer-wise operators for the ReduNet [7].

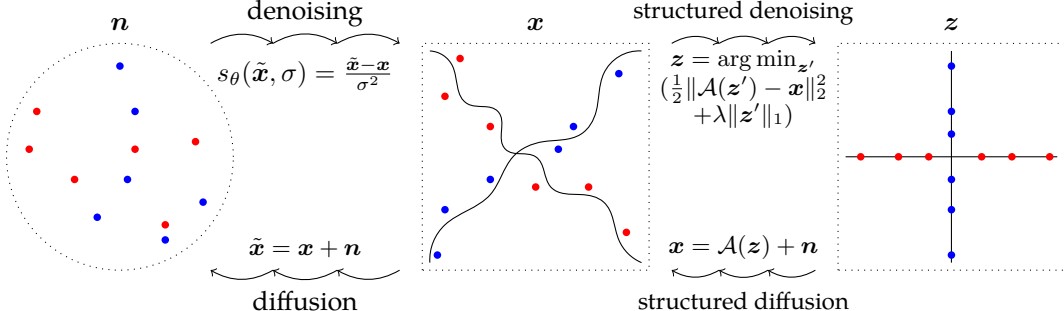

Figure 7: Connections and contrasts between "traditional" diffusion and our structured diffusion/denoising processes. While conventional diffusion and denoising process consider isotropic noise, our process consider generation and denoising against a learned (convolutional) dictionary $\mathcal{A}$. The goal is to obtain a more, compact, structured (e.g. sparse) internal representation $z$.

## B. Ablation Study on Optimization Strategies

In this section, we justify our choice of optimization strategy to optimize equation 10. We set the following optimization strategy as "Strategy 1", which was adopted in the original CTRL:

$$\max_{\theta(\boldsymbol{A})} \Delta R \text{ step}: \quad \boldsymbol{A}_{k+1} = \boldsymbol{A}_k + \lambda_{\max} \frac{\partial \Delta R}{\partial \theta} \cdot \frac{\partial \theta}{\partial \boldsymbol{A}} \Big|_{\boldsymbol{A}_k}, \tag{13}$$

$$\min_{\eta(\boldsymbol{A})} \Delta R \text{ step}: \quad \boldsymbol{A}_{k+2} = \boldsymbol{A}_{k+1} - \lambda_{\min} \frac{\partial \Delta R}{\partial \eta} \cdot \frac{\partial \eta}{\partial \boldsymbol{A}} \Big|_{\boldsymbol{A}_{k+1}}. \tag{14}$$

We set the following optimization strategy as "Strategy 2", which is used in our work to optimize equation 10.

$$\max_{\theta(\boldsymbol{A})} \Delta R \text{ step}: \quad \boldsymbol{A}_{k+1} = \boldsymbol{A}_k + \lambda_{\max} \frac{\partial \Delta R}{\partial \theta} \cdot \frac{\partial \theta}{\partial \boldsymbol{A}} \Big|_{\boldsymbol{A}_k}, \tag{15}$$

$$\min_{\boldsymbol{A}} \Delta R \text{ step}: \quad \boldsymbol{A}_{k+2} = \boldsymbol{A}_{k+1} - \lambda_{\min} \Big( \frac{\partial \Delta R}{\partial \eta} \cdot \frac{\partial \eta}{\partial \boldsymbol{A}} + \frac{\partial \Delta R}{\partial \theta} \cdot \frac{\partial \theta}{\partial \boldsymbol{A}} \Big) \Big|_{\boldsymbol{A}_{k+1}}. \tag{16}$$

We run an ablation study on CIFAR-10 with hyper-parameters all the same from Appendix A.1, except the training strategy. The results are shown in Table 6. Empirically, we found that Strategy 2 optimizes much better than Strategy 1.

|  | IS ($\uparrow$) | FID ($\downarrow$) |
|---|---|---|
| Strategy 1 | 3.2 | 197.1 |
| Strategy 2 | 8.9 | 28.9 |

Table 6: Ablation study of CSC-CTRL on different optimization strategies through reconstructed image quality (IS/FID). $\uparrow$ means the higher the better. $\downarrow$ means the lower the better.

## C. More Visualization of CSC-CTRL Generated Images

Due to limited space in the main body, we show the generated images of STL-10 (see Figure 8) and some extra images of ImageNet (Figure 9) in this section. Figure 8 shows the auto-encoding properties of our learned framework on STL-10. Figure 9 shows a larger version of the reconstruction on ImageNet. We observe that even fine details in the image have been faithfully reconstructed, showcasing the power of our convolutional sparse coding network. Lastly, we include more generated images on ImageNet in Figure 10, demonstrating the image quality of our network.

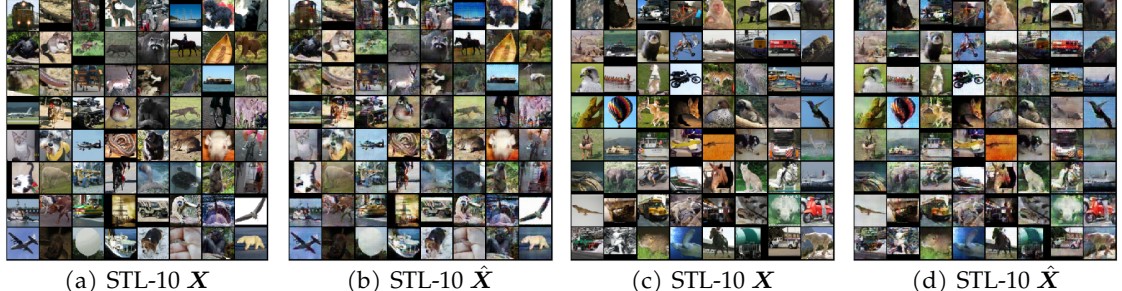

| (a) STL-10 $\boldsymbol{X}$ | (b) STL-10 $\hat{\boldsymbol{X}}$ | (c) STL-10 $\boldsymbol{X}$ | (d) STL-10 $\hat{\boldsymbol{X}}$ |

Figure 8: Visualizing the auto-encoding property of the learned CSC-CTRL ($\hat{\boldsymbol{X}} = g(f(\boldsymbol{X}, \theta), \eta)$) on STL-10. (Images are randomly chosen.)

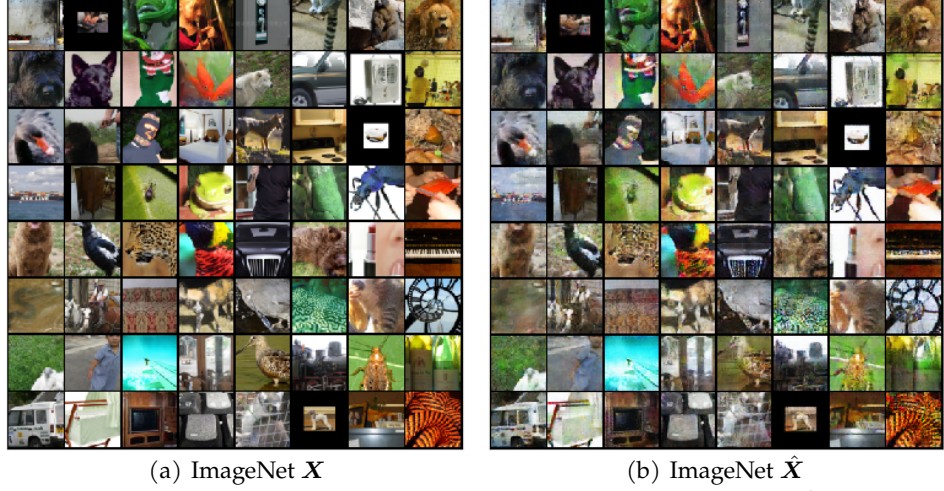

| (a) ImageNet $\boldsymbol{X}$ | (b) ImageNet $\hat{\boldsymbol{X}}$ |

Figure 9: Visualizing the auto-encoding property of the learned CSC-CTRL ($\hat{\boldsymbol{X}} = g(f(\boldsymbol{X}, \theta), \eta)$) on ImageNet. (Images are randomly chosen.)

## D. Learned Structured Feature Space

**Linear interpolation.** Figure 11 shows reconstructed images whose features are linearly interpolated between pairs of images sampled from the class of "beach wagon" of ImageNet dataset (the class ID: n02814533). Formally, for two images $\boldsymbol{x}_1, \boldsymbol{x}_2$, the interpolated $\boldsymbol{x}$ is given by

$$\boldsymbol{x}_{\text{interp}} = g(\alpha f(\boldsymbol{x}_1) + (1 - \alpha) f(\boldsymbol{x}_2)) \tag{17}$$

where $\alpha \in [0, 1]$ varies in Figure 11 from $0$ (on the left side) to $1$ (on the right side).

The generated images show a continuous deformation from one sample to another. This verifies that our feature space is linearized and discriminative.

## E. More Analysis of Denoising

### E.1. Quantitative Measure of Image Denoising Quality

Due to space limitations in the main body, we present a quantitative analysis of denoising in this section. We use PSNR (Peak Signal-to-Noise Ratio), MSE (Mean Squared Error) and SSIM (Structural Similarity Index Measure) to measure the quality of denoising via CTRL and CSC-CTRL. Shown in Table 7, CSC-CTRL performs significantly better than CTRL trained with the usual convolutional layers. It quantitatively verifies the effectiveness of the convolutional sparse coding layer for denoising.

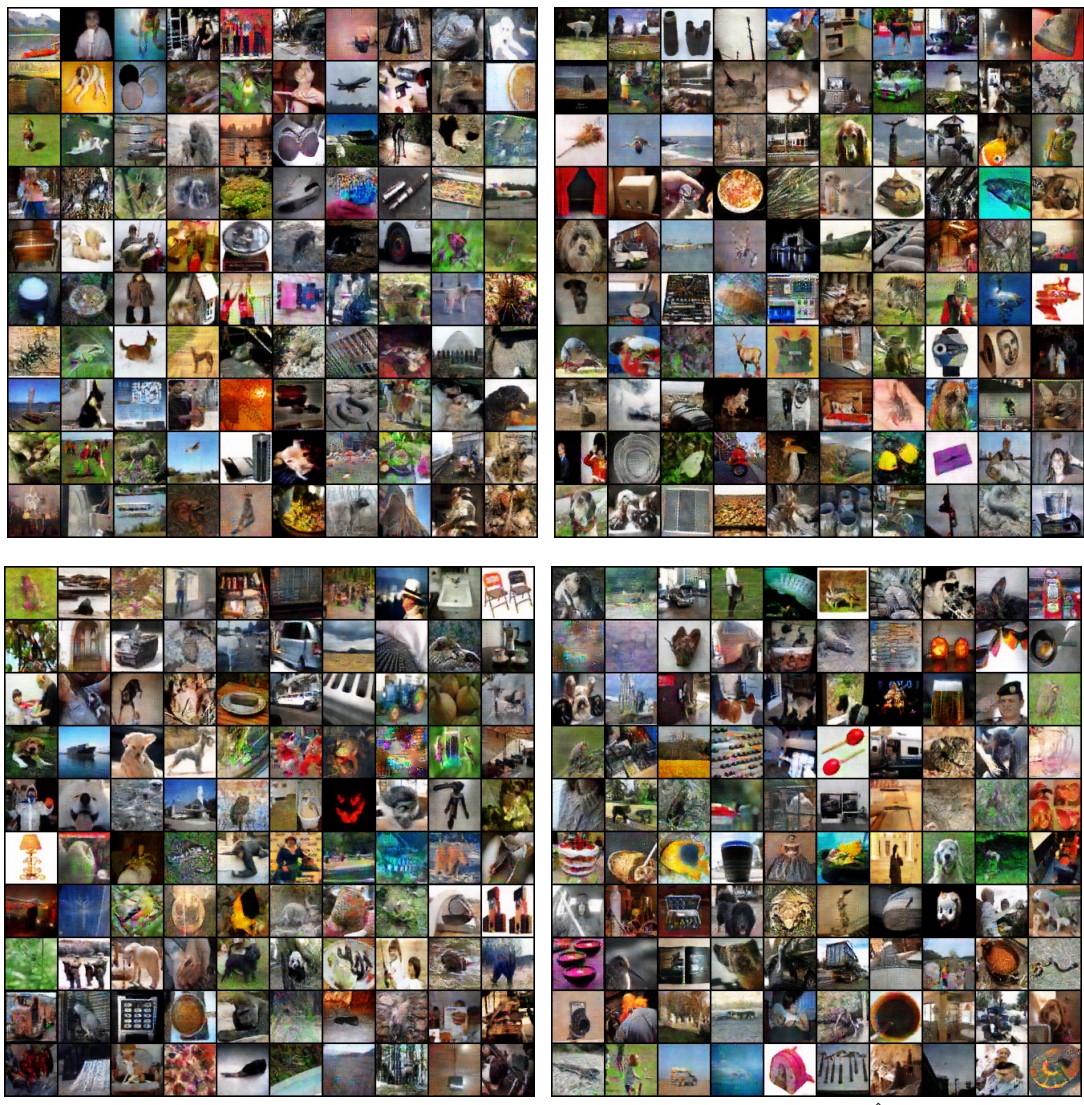

Figure 10: Visualizing randomly chosen reconstructed images of CSC-CTRL ($\hat{\boldsymbol{X}} = g(f(\boldsymbol{X}, \theta), \eta)$) on ImageNet.

| Noise level ($\sigma = 0.5$) | PSNR ($\uparrow$) | MSE ($\downarrow$) | SSIM ($\uparrow$) |
|---|---|---|---|
| CTRL | 13.3961 | 0.1914 | 0.1556 |
| CSC-CTRL | 17.0938 | 0.0837 | 0.3671 |

Table 7: Comparison of denoising via CTRL and CSC-CTRL with standard metrics. $\uparrow$ means the higher the better. $\downarrow$ means the lower the better.

## E.2. Better Denoising through Adjusting Sparse Factor

In fact, we can get better denoising effect by simply adjusting the $\lambda$ in the convolutional sparse coding layer in equation 6 **without** any additional training. Our default $\lambda$ is set to be $0.01$ due to the scale between two objectives during the training stage. In the inference stage, we can further increase $\lambda$ to promote sparsity, which naturally leads to better denoising. From Table 8, we see that as $\lambda$ increases, CSC-CTRL generally improves at denoising.

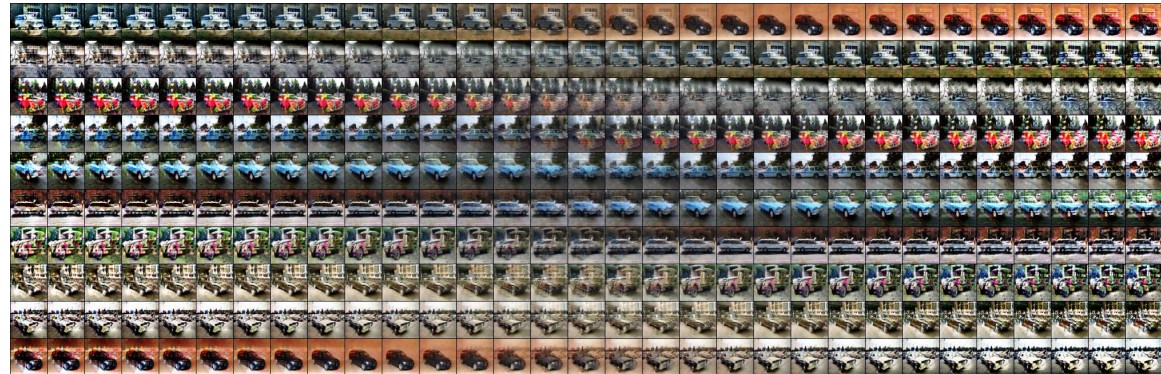

Figure 11: Images generated by features which were linearly interpolated between pairs of images sampled from the class of "beach wagon" of ImageNet dataset (the class ID: n02814533) in the learned feature space.

| Noise level ($\sigma = 0.5$) | PSNR ($\uparrow$) | MSE ($\downarrow$) | SSIM ($\uparrow$) |
|---|---|---|---|
| $\lambda = 0.01$ (default) | 17.0938 | 0.0837 | 0.3671 |
| $\lambda = 0.1$ | 17.5774 | 0.0733 | 0.3955 |
| $\lambda = 0.2$ | 17.9926 | 0.0655 | 0.4222 |
| $\lambda = 0.3$ | 18.3500 | 0.0602 | 0.4479 |
| $\lambda = 0.4$ | 18.6068 | 0.0572 | 0.4658 |
| $\lambda = 0.5$ | **18.6155** | **0.0567** | **0.4676** |
| $\lambda = 0.6$ | 18.4205 | 0.0601 | 0.4593 |
| $\lambda = 0.7$ | 18.0563 | 0.0660 | 0.4364 |

Table 8: Comparison of denoising using different $\lambda$ with standard metrics. $\uparrow$ means the higher the better. $\downarrow$ means the lower the better.

## F. Stability

In this section, we further verify the training stability of CSC-CTRL from two perspectives: mode collapse during training, and choice of batch size.

**Training Stability.** Experimentally, many previous methods such as CTRL and various GANs suffer from training instability. As shown in Figure 12, CTRL shows a clear training instability after 600 epochs. In contrast, CSC-CTRL training is much more stable, as the IS score barely drops. We conclude that CSC-CTRL suffers less from mode collapse.

**Choice of Batch Size.** One notable flaw of the original CTRL [12] is its reliance on a large batch size, normally greater than 512. This large batch size greatly increases the model's computational cost and limits its scalability. In Table 9, we compare whether each method converges under different batch sizes, from as small as 10 to as large as 2048. From the table, we see that CSC-CTRL can successfully converge on a wider range of batch sizes, even as low as 10. This greatly reduces the required computation power and enables easier training on more complicated datasets such as ImageNet.

| Batch Size | 10 | 64 | 128 | 256 | 512 | 1024 | 1600 |
|---|---|---|---|---|---|---|---|
| CSC-CTRL | ✓ | ✓ | ✓ | ✓ | ✓ | ✓ | ✓ |
| CTRL | ✗ | ✗ | ✗ | ✗ | ✓ | ✓ | ✓ |

Table 9: Comparison of CTRL and CSC-CTRL trained with different batch sizes. ✓ means the method has successfully converged, ✗ means the method fails to converge.

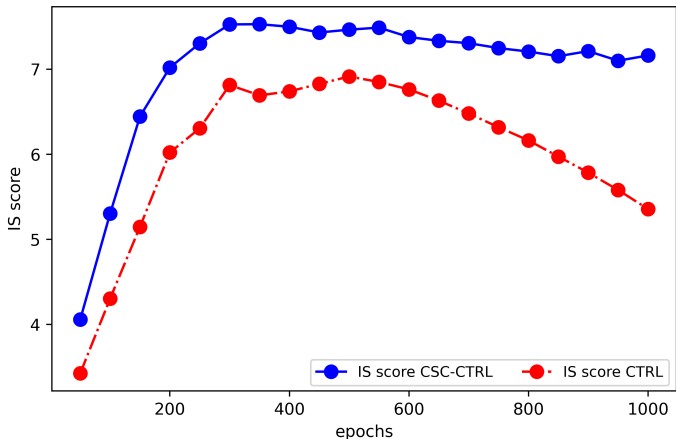

Figure 12: Training stability comparison of CTRL and CSC-CTRL with IS score on CIFAR-10.

| Method | Model Size | Train Time | CIFAR-10 | | STL-10 | | ImageNet | |
|---|---|---|---|---|---|---|---|---|
| | | | IS↑ | FID↓ | IS↑ | FID↓ | IS↑ | FID↓ |
| *GAN based methods* | | | | | | | | |
| DCGAN [50] | | | 6.6 | 35.3 | 7.8 | - | - | - |
| SNGAN [43] | | | 7.4 | 29.3 | 9.1 | 40.1 | 7.3 | 48.7 |
| *VAE based methods* | | | | | | | | |
| VAE [29] | | | 5.2 | 55.9 | - | - | - | - |
| NVAE [63] | 10M | >55 h | - | 50.8 | - | - | - | - |
| NVAE (Recon) | 10M | >55 h | - | 2.67 | - | - | - | - |
| DCVAE [48] | 4M | >24h | 8.2 | 17.9 | 8.1 | 41.9 | - | - |
| DCVAE (Recon) | 4M | >24h | 7.9 | 21.4 | 8.4 | 43.6 | - | - |
| *Flow based methods* | | | | | | | | |
| GLOW [30] | | | - | 46.9 | - | - | - | - |
| Residual Flow [8] | | | - | 50.8 | - | - | - | - |
| *CTRL based methods* | | | | | | | | |
| CTRL [12] | 1.0M | 15 h | 8.1 | 19.6 | 8.4 | 38.6 | 7.7 | 46.9 |
| CSC-CTRL (ours) | 0.5M | 8 h | 8.9 | 28.9 | 9.1 | 48.1 | 12.5 | 34.5 |

Table 10: Comparison on CIFAR-10, STL-10, and ImageNet-1K. The network architectures used in CSC-CTRL are 4-layers for CIFAR-10, 5-layers for STL-10 and ImageNet respectively which are much smaller than other compared methods. NVAE(recon) means the results of reconstruction, the column "Train Time" means the hours the model used for training.

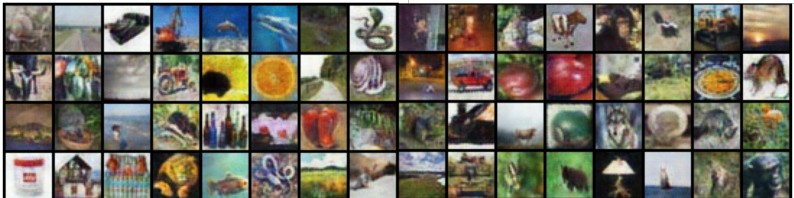

Figure 13: Visualization of randomly chosen reconstructed samples $\hat{X}$ of CIFAR-100. The autoencoding model is only trained on the CIFAR-10 dataset.

