# OpenReview forum: "Closed-Loop Transcription via Convolutional Sparse Coding"
_CPAL.cc/2024/Conference — CPAL 2024 (Proceedings Track) Oral_

### Official Review · Reviewer_jNki · 2023-10-07
**Well-written paper with convincing experiments**

**Rating:** 6
**Confidence:** 4

**Review:**

This paper introduces a new autoencoder design that uses convolutional sparse coding (CSC) and trains it using the closed-loop transcription (CTRL) method. The resulting generative model performs well in both image reconstruction and generation.

Pros:
1. Clear writing: The paper is easy to understand, and the experiments are well explained and supported.
2. Good literature review: The related works section covers the most relevant research in the field.
3. Strong experiments: The experiments are convincing, with comparisons to other models that make the proposed approach look solid.

Cons:
1. Discuss weaknesses: The paper should discuss its limitations more clearly.
2. Higher resolution experiments: Please test the model on higher-resolution datasets like 128x128 if possible, or explain why not and how it compares to the baselines in this regard.
3. Include timing information: Please clarify how long the training and inference take compared to the baselines.
4. Reconstruction quality metrics: Please use metrics like PSNR or SSIM to measure image quality in Figures 2, 3, and 5.
5. Please show the number of trainable parameters for different models in Table 1. You can also add training and inference times to this table.
6. Please Include the original images in Figure 5 so readers can see how well the model recreates them.
7. Please make Figure 4 clearer by adding more space between blocks and brief captions for each block.

I believe this is overall a decent paper. The proposed framework is promising but should address these concerns to improve its overall quality and clarity. These improvements could lead to a higher evaluation score.

---

### Official Review · Reviewer_iEnQ · 2023-10-07
**Important direction and a well-written paper. The method is principled but may lack practicality.**

**Rating:** 6
**Confidence:** 2

**Review:**

Summary:

This paper proposes an auto-encoding architecture that implements convolutional sparse coding layers with weighting sharing between the encoder and the decoder. Trained with CTRL, the model exhibits decent scalability, structured representation, and superior sample-wise alignment.

Merits:

1. The paper is well written, carefully introducing and discussing technical details.

2. The direction toward more powerful white-box models is relevant and essential in the era of large black-box models.

3. The proposed approach is well-grounded with principles. Each component has a clear interpretation, including the convolutional kernels shared by the encoder, the decoder, and the CTRL objective.

4.  The method exhibits stronger scalability and reconstruction performance than its precursor CTRL.


Disadvantages:

1. I am unsure about the experimental comparison with popular deep-learning generative models like VAEs and GANs. If I understand it correctly, the proposed approach can also perform reconstruction, whereas VAEs and GANs are designed to sample novel samples from the training distribution. Therefore, the comparison in Table 1 seems unfair (or not comparable) to me. Plus, the lack of the capability of sampling novel samples is undesirable in comparison.

2. The encoding process involves iterative algorithms, which may significantly slow the training and induce numerical instability. On this aspect, it would be appreciated if the authors could offer training time reports contrasting the evaluated approaches.

---

### Official Review · Reviewer_BBBs · 2023-10-08
**Incorporating convolutional sparse coding in CTRL generative framework**

**Rating:** 6
**Confidence:** 3

**Review:**

**Summary of the proposed work:**

Conventional autoencoders typically use generic network architectures and lack structure or interpretability in the latent code representations. A recently proposed CTRL framework uses rate reduction to build generative models while imposing a Gaussian mixture model like structure in the latent codes.

The submitted work proposes CSC-CTRL in which the generation of natural images is modeled using a stacked concatenation of multiple convolutional sparse coding (CSC) layers and incorporated within the overall CTRL framework. The atoms in the convolutional dictionaries of the encoder and decoder are kept the same which allows for sample-wise alignment in the generated samples—an improvement over existing CTRL approach.

The paper shows good image generation quality with interpretability in latent codes with respect to image classes in the dataset. There is additionally a higher stability to noise than vanilla CTRL owing to the incorporation of sparsity.

**Pros:**

- The results are nice and convincing with respect to the main ideas in the paper’s body.
- Barring a few minor typos, the paper is generally clearly written.

**Cons:**

I have the following questions/concerns that I would like the authors to clarify.

- The authors emphasize that the key goal is to show the feasibility of high quality image generation using CSC layers. While their results are interesting, there exist a few recent works which also use multi-scale CSC to generate high quality images, for example [1] and [2].  These focus on imaging inverse problems rather than autoencoding, though one could extend them to autoencoding by say tying the weights of their encoders and decoders.

There are indeed some differences between the proposed work and the 2 references, for example, [1] performs direct pixel level MSE loss minimization instead of distribution matching or CTRL. However, given similarities like CSC based high quality generation and interpretability in obtained dictionaries, a comparison (conceptual or experimental) would be helpful to better place the contributions of the submission with respect to literature.

- One of the advantages mentioned is that CSC-CTRL has reduced computational cost over prior models (line 48 in the paper). But wouldn’t the use of FISTA iterations in the encoder’s forward pass add extra overhead making it more expensive than a conventional autoencoder?


[1] Liu, Tianlin, et al. "Learning multiscale convolutional dictionaries for image reconstruction." IEEE Transactions on Computational Imaging 8 (2022): 425-437

[2] Li, Minghan, et al. "Video rain streak removal by multiscale convolutional sparse coding." Proceedings of the IEEE conference on computer vision and pattern recognition. 2018.

---

### Meta-Review · Area_Chair_RcdN · 2023-11-14

**Recommendation:** Accept (Poster)
**Confidence:** 5

**Metareview:**

The CSC-CTRL framework proposed by the authors is an interesting combination of convolutional sparse coding and a distributional loss function related to compression. The resulting autoencoders are more lightweight than the compared convolutional networks, while yielding perceptually good quality images and outperforming some of the comparable generative techniques proposed earlier. Overall it is intriguing to see that this convolutional framework can perform so well and yield strong sample-level encoding.

For balance, I also want to state some places where the manuscript could be improved. The "interpretability" aspect is in my opinion overstated. I don't think that the sparse feature representations are any more or less interpretable than feature maps in convnets, nor is the related generative process more interpretable than a convnet decoder. Since 2 iterations of FISTA are generally not enough to produce the _sparsest_ code, the feature maps here are _some_ sparse feature maps which are optimized end-to-end for generative performance.

Some of the arguments in responses seem to sidestep important issues—for example, why it is not necessary to compare with SOTA. Naming "engineering tricks" as culprits for performance differences sounds somewhat evasive, especially without a detailed elaboration, and acknowledging that the authors work with tiny images. Batch normalization and ReLUs in between the "principled" (de)convolution layers could similarly be declared engineering tricks (without a deeper discussion, at least). Finally, I doubt that CSC strategies can scale to high resolutions without explicitly accounting for regularity across scales and directions, which necessarily involves some kind of multiscale design. But it would be intriguing to be proven wrong.

I agree with the reviewers that this paper is marginally above the acceptance threshold and urge the authors to incorporate the promised changes, including improved comparisons with prior art, in the camera ready version.

---

### Decision · Program_Chairs · 2023-11-19

**Decision:**

Accept (Oral)

**Comment:**

The paper presents the CSC-CTRL framework, which combines convolutional sparse coding (CSC) with the CTRL framework for image generation. Reviewers generally find the paper's contributions promising, with good image generation quality and interpretability in latent codes. However, they raise several concerns and suggestions for improvement, including the need for comparisons with related works that use multi-scale CSC for image generation, clarification on computational costs, and the inclusion of higher-resolution experiments. Reviewers also request additional information such as training times, reconstruction quality metrics, and clearer figures. Overall, the paper has potential but requires some refinements and clarifications to address these concerns.

The action PC chair for this paper is Atlas Wang, who made the decision after carefully reading the paper as well as the comments by all reviewers and AC. The decision is agreed by all PC chairs.